# Safety, Feasibility and Efficacy of Lokomat^®^ and Armeo^®^Spring Training in Deconditioned Paediatric, Adolescent and Young Adult Cancer Patients

**DOI:** 10.3390/cancers15041250

**Published:** 2023-02-16

**Authors:** Morgan Atkinson, Angela Tully, Carol A. Maher, Christopher Innes-Wong, Ray N. Russo, Michael P. Osborn

**Affiliations:** 1Youth Cancer Service SA/NT, Royal Adelaide Hospital, Adelaide 5000, Australia; 2Paediatric Rehabilitation Department, Women’s and Children’s Hospital, Adelaide 5006, Australia; 3Alliance for Research in Exercise, Nutrition and Activity (ARENA), Allied Health and Human Performance, University of South Australia, Adelaide 5000, Australia; 4School of Medicine, Flinders University, Adelaide 5042, Australia; 5Department of Haematology and Oncology, Women’s and Children’s Hospital, Adelaide 5006, Australia; 6Adelaide Medical School, University of Adelaide, Adelaide 5000, Australia

**Keywords:** adolescent and young adult, robot-based therapy, deconditioned, child, exercise, VO_2peak_, cancer

## Abstract

**Simple Summary:**

Robot-based training is a field of study which aims to understand and augment rehabilitation through the use of robotic orthotics. This study aimed to assess the safety, feasibility and efficacy of rehabilitation robotics in children, adolescents and young adults (AYAs) during or soon after cancer treatment. Cancer patients with significant musculoskeletal, neurological, gait and/or upper limb deficiencies were recruited. Robotic devices utilised real-time biofeedback and computer games to engage participants. 76% completed the 6-week intervention with an overall adherence of 83%. The mean participant satisfaction score was 8.8/10. Fourty-nine adverse events were recorded throughout the course of the study, and no adverse event led to withdrawal from the study. Results suggest that robot-based rehabilitation is safe and feasible for use in children and AYAs who are currently undergoing or recently completed cancer therapy. Preliminary efficacy results indicate large beneficial effects on functional and patient-reported outcomes.

**Abstract:**

Background: Rehabilitation robotics is a field of study which aims to understand and augment rehabilitation through the use of robotics devices. Objective: This proof of concept study aimed to test the safety (no. adverse events, incidence of infection), feasibility (program demand, adherence, participant satisfaction) and efficacy (Peak Oxygen uptake (VO_2peak_), 6-min walk test, gait speeds, Canadian Occupational Performance Measure, quality of life) of Lokomat^®^ and Armeo^®^Spring training in children and adolescents and young adults (AYAs) during or soon after cancer treatment. Method: This was a 6-week single arm pre-post study. Cancer patients with significant musculoskeletal, neurological, gait and/or upper limb deficiency aged 5 to 25 years were recruited. The rehabilitation program included access to two robotic orthoses: the Lokomat^®^ and/or Armeo^®^Spring. Robotic devices utilised real-time biofeedback and computer games to engage and guide participants through a repetitive functional range of movement aimed at improving functional deficiencies. Progressive increases in exercise intensity and duration were encouraged. Results: Twentey-eight participants were approached for study; twenty-one consented. Seventy-six percent completed the six-week intervention with an overall adherence of 83%. The mean participant satisfaction score was 8.8/10. Forty-nine adverse events were recorded throughout the course of the study, forty-five grade 1, three grade 2 and one grade 3. No adverse events led to withdrawal from the study. Preliminary efficacy results indicate large beneficial effects on VO_2peak_ (r = 0.63), 10 m comfortable pace walk (r = 0.51) and maximal pace walk (r = 0.60), 6-min walk test (r = 0.60), maximal back and leg strength (r = 0.71), trunk flexibility (r = 0.60), The European Organization for the Research and Treatment of Cancer Quality of Life Questionnaire (EORTC QLQ C30) (r = 0.61), Functional Assessment of Chronic Illness Therapy–Fatigue (FACIT F) r = 0.53 and the Canadian Occupational Performance Measure, satisfaction (r = 0.88) and performance scores (r = 0.83), and moderate beneficial effects on Leisure Score Index (LSI) (r = 0.30). Conclusion: Our results suggest that Lokomat^®^ and Armeo^®^Spring training is safe and feasible for use in children and AYAs who are currently undergoing or have recently completed cancer therapy. A larger controlled trial investigating the efficacy of robotics rehabilitation in this cohort is warranted.

## 1. Introduction

Cancer treatment in children, adolescents and young adults (AYAs) is associated with impairments in cardiovascular fitness and functional deficits. These may persist into survivorship and are associated with fatigue, impaired quality of life, cardiovascular disease risk and late mortality [1,2,3,4,5,6,7,8]. This loss of function is likely to be a result of the direct toxic effects of anticancer therapy as well as the indirect consequences of therapy such as sedentary behaviours. [1,5,6,7,8,9,10,11]. This commonly manifests as fatigue and impaired quality of life in cancer survivors.

Over the past decade, there has been increased research and clinical interest in the role of exercise training following a cancer diagnosis. Studies in children, AYAs and older adults with cancer have demonstrated that structured exercise is safe (during and after cancer therapy), well tolerated (adherence rates > 80%), and associated with improvements in cardiorespiratory fitness, muscle strength, and patient-reported outcomes, including but not limited to quality of life, fatigue and depression [1,12,13,14,15].

Despite the promising results obtained from exercise programmes, some patients are incapable of participating due to factors such as recent limb surgery (e.g., bone and soft tissue sarcomas), incoordination or hemiparesis (e.g., brain tumours), severe peripheral neuropathy following vincristine treatment, or proximal myopathy following corticosteroids [1]. Furthermore, premorbid exercise habits and issues relating to body image may prevent some patients from engaging in structured exercise programs [16]. Novel approaches are required for such patients, which consider their post-surgical, neurological, or musculoskeletal deficits, body image, and premorbid exercise frequency and intensity.

Rehabilitation robots are emerging as a promising therapeutic tool for children and AYAs with loss of mobility and disability [17,18,19,20,21]. They help to restore function and mobility and can be classified by their mechanical structure, i.e., end effectors or exoskeletons [17]. End effectors apply forces to the distal segments of limbs, creating a “mechanical chain” that prompts other parts of the limb to move [17]. In contrast, exoskeletons are “wearable machines” that are adjusted to the patient’s skeletal structure and move the joint of the limb where the exoskeleton is worn [17].

There are multiple upper and lower limb rehabilitation robotic devices commercially available, and/or in proof of concept [17]. The Lokomat^®^ (Hocoma AG, Volketswil, Switzerland) is a commercially available lower limb robotic exoskeleton with a body weight support system used in parallel with a treadmill that replicates lower limb biomechanics and interfaces with an augmented virtual reality system. Linear electrical motors help drive the hip and knee while the foot lift induces passive dorsiflexion of the ankle [17]. Armeo^®^Spring (Hocoma AG, Volketswil, Switzerland) is a commercially available upper limb exoskeleton that supports the participant’s arm and includes an adjustable suspension system which connects with game-based virtual reality with varying degrees of complexity [22]. The exoskeleton can be calibrated to the patient’s active range of movement and provides information about specific movement parameters such as resistance, strength, range of motion and coordination [22].

The use of rehabilitation robots such as the Lokomat^®^ and Armeo^®^Spring can have advantages over conventional physiotherapy, as they allow extensive and precise repetition of movement in patients with loss of mobility or disability, require reduced manual handling for clinicians administering the therapy, and provide quantitative feedback on range of motion and strength with each repetition. Moreover, their ability to interface with game-based virtual reality may also improve patient motivation and engagement [17,22].

Studies investigating both the Lokomat^®^ and Armeo^®^Spring have proved effective and well-tolerated in children and AYAs with neuromuscular conditions, including stroke, children with cerebral palsy, traumatic brain injuries and following orthopaedic procedures [18,23,24,25,26]. Potential benefits include improvements in gait speed (10 MWT) and endurance (6 MWT), joint range of movement (upper and lower limb), fatigue and functional performance measures such as the Canadian Occupational Performance Measure. While most studies investigating Lokomat^®^ and Armeo^®^Spring deem them to be effective, they are generally not superior to conventional therapies [27]. There is also a need for greater clarity for clinicians to safely apply the technology [23], as little is known about the causality of adverse events and the tolerability and acceptability of applying this technology [26].

Despite the progressive development of rehabilitation robotic devices and robot-based therapy, their application to the paediatric and AYA population is still scarce. Children and AYAs with cancer suffer from a loss of mobility and disability which can limit activities of daily living and compromise quality of life well into survivorship [28]. However, to our knowledge robot-based therapies such as the Lokomat^®^ and Armeo^®^Spring have not been investigated in children and AYAs with cancer. Therefore, the primary aim of this study was to determine the safety and feasibility of Lokomat^®^ and Armeo^®^Spring training in children and AYAs who were undergoing or recently completed cancer treatment. The secondary aim of this study was to determine whether Lokomat^®^ and Armeo^®^Spring improved exercise and functional capacity, daily performance, quality of life and fatigue.

## 2. Materials and Methods

This was a single-arm pre-post feasibility study. Multisite ethics approval was obtained from the Women’s and Children’s Health Network Human Research Ethics Committee (HREC/17/WCHN/175). Written informed consent was obtained from all participants, with data collection taking place between May 2019 and June 2021. This manuscript was prepared in line with CONSORT feasibility trial guidelines [29] (pp. 5–13), and the intervention is described according to the TIDIER checklist [30].

### 2.1. Participants

Potential participants were identified by treating physicians or via the Youth Cancer Service South Australia/Northern Territory state-wide multidisciplinary team meeting, and approached during oncology outpatient appointments or during an inpatient admission at the Women’s and Children’s Hospital or Royal Adelaide Hospital (both in Adelaide, Australia). Patients were eligible if they were diagnosed with a hematological malignancy or solid tumour, aged 5–25 years, currently undergoing or completed cancer treatment, were experiencing a movement impairment relating to cancer treatment (significant musculoskeletal, neurological, gait or upper limb deficiency) and had written medical clearance to participate. Participants were excluded from the study if they had absolute contraindications to exercise e.g., unstable angina, uncontrolled heart failure, acute systemic infection accompanied by fever, cognitive impairment severe enough to limit participation in the rehabilitation robotics intervention (as determined by a medical practitioner), an inability to convey discomfort and the need to stop therapy, and had <6 months life expectancy. Lokomat-specific exclusion criteria included height < 125 cm, Femur length < 21 cm, and body mass < 15 kg (required to use the Lokomat), and presence of neurological dysfunction such as reduced sensation and contractures.

### 2.2. Intervention

Robot-based training sessions were conducted in the Women’s and Children’s Health Network’s Little Heroes Foundation Centre for Robotics and Innovation. Two robotic training devices were available. The Lokomat^®^ (Lokomat^®^Pro, Hocoma, Switzerland) is a robotic gait orthosis designed to improve walking pattern and function. Lokomat^®^ progressions included decreasing body weight support (increasing load), increasing gait speed, and reducing guidance force. The Armeo^®^Spring (Armeo^®^Spring, Hocoma, Switzerland) is a robotic upper limb orthosis designed to improve upper limb muscle strength, range, coordination and function. Armeo^®^Spring progressions included reducing limb weight support (increasing load), reducing motor control support, increasing task speed and moving from 2D to 3D activities.

Participants were able to enrol on both the Armeo^®^Spring and/or Lokomat^®^ if clinically indicated; however, they were not able to complete Armeo^®^Spring or Lokomat^®^ during the same 6-week period. Participants were asked to attend two training sessions per week for 6 weeks. Sessions were overseen by an accredited exercise physiologist (AEP) with experience in oncology, and a physiotherapist with specific Lokomat^®^ and Armeo^®^Spring training. Prior to each session, participants (or their caregivers) completed a safety questionnaire to identify any absolute or relative contraindications to exercise or Lokomat^®^/Armeo^®^Spring intervention (Appendix A—Safety Checklist). If absolute or relative contraindications to exercise or robotics intervention were identified, an oncologist was consulted to determine whether it was safe to proceed.

Interventions were tailored to each participant with the aim of addressing specific functional limitations such as gait or upper limb deficiency. Participants completed a minimum of 20 min on either the Lokomat^®^ or Armeo^®^Spring with progressive increases in duration to a maximum of 40 min. Robot settings (decreasing body weight support, increasing gait speed, reducing guidance force, reducing upper limb weight support, reducing motor control support and increasing task speed) were individually adjusted according to the participant’s abilities and goals, and progressed at the clinicians’ discretion, based on exercise prescription, duration, and progression principles recommended by the American College of Sports Medicine and Exercise and Sports Science Australia [4,31]. Heartrate and SpO_2_ was monitored using a Nellcor^TM^ forehead SpO_2_ sensor (Medtronic, Boulder, CO, USA) and used to progress exercise intensity along with verbal feedback from the participants. If a participant was feeling unwell or uncomfortable, the session was either trialled at a lower intensity or stopped.

### 2.3. Outcomes

Demographic information collected at baseline included: age, sex, education, cancer diagnosis, smoking status, presence of peripheral neuropathy, proximal myopathy, or other neurological or musculoskeletal signs on physical examination.

### 2.4. Safety

Adverse events, occurring before, during or after each training session, were recorded using the Common Terminology Criteria for Adverse Events v5.0 tool [32]. This tool allows treating clinicians to grade the severity (Grade 1 Mild, Grade 2 Moderate, Grade 3 Severe, Grade 4 Life-threatening, Grade 5 Death) and likely causality (Certain, Probably, Possible, Unlikely) of events.

### 2.5. Feasibility

Feasibility was measured via program demand, adherence, and patient/family perceptions of the intervention. Program demand was assessed via the number of participants approached versus the number who consented to the study. The number of people excluded and the reasons for exclusion were recorded. Adherence (the number of robot-based therapy sessions attended, divided by the number of sessions prescribed) was calculated. A purpose-designed survey completed after the final training session was used to collect subjective information on the feasibility of attending a Lokomat^®^ or Armeo^®^Spring rehabilitation program for participants and their families (Appendix A—Robotics Questionnaire). It included eight questions ranked on a scale from strongly disagree to strongly agree (e.g., “I would continue with sessions if they were available”) and a section to capture open-ended comments regarding the benefits and drawbacks of the intervention. Clinician field notes were documented during each Lokomat^®^ or Armeo^®^Spring training session to further inform safety and feasibility, including mechanical or technical faults relating to the robotics equipment.

### 2.6. Preliminary Efficacy

Participants completed functional assessments at baseline and 6 weeks ( ±2 weeks).

For participants who used the Lokomat^®^, the following outcomes were assessed:Cardiopulmonary fitness was assessed using cardiopulmonary fitness tests on a cycle-ergometer and conducted according to standardised procedures (American Thoracic Society/American College of Chest Physicians) [33].Walking endurance was assessed using the six-minute walk test [34].Gait efficiency was assessed using the ten-metre walk test (comfortable pace and fast pace) [35].Maximal back and leg strength were assessed using a hydraulic back and leg dynamometer (Baseline Back-Leg-Chest Dynamometer, Via Industrial, Bogotá, DC Colombia) [36,37].Trunk flexibility was measured using a sit and reach box (Flex-Tester Sit and Reach Flexibility Test Box, Novel Products Inc., Rockton, IL, USA) [38].

For participants who used the Armeo^®^Spring, the following outcomes were assessed:
Unilateral gross manual dexterity was assessed using the box and block test [39].Shoulder flexibility was assessed using the back scratch test [39].Maximal grip strength was assessed using a hydraulic hand grip dynamometer (Saehan, SH5001, Masan, Republic of Korea) [40].Muscular endurance was assessed using the 30-s arm curl test (5lb and 8lb) [41].In addition, all participants completed the following self-reported questionnaires at baseline and 6 weeks:Quality of life was assessed using the European Organization for the Research and Treatment of Cancer Quality of Life Questionnaire (EORTC QLQ-C30) [42].Fatigue was assessed by the Functional Assessment of Chronic Illness Therapy-Fatigue (FACIT-F) [43].Leisure time physical activity was assessed using the Godin-Shephard Leisure Time Physical Activity Questionnaire (GSLTPAQ) [44].Activities of daily living were assessed using the Canadian Occupational Performance Measure (COPM) [45].

### 2.7. Sample Size

As this was a safety and feasibility study, a sample size calculation was not performed. Based on annual data regarding patient numbers and the intensive nature of the intervention (and therefore budgetary constraints), the a priori target sample size was 20 participants.

### 2.8. Statistical Analysis

Data were analysed using SPSS (version 27, IBM, Armonk, NY, USA). Participant characteristics and safety and feasibility data were analysed descriptively, using means, standard deviations (or medians and interquartile ranges) for continuous data, and the count and percentages for categorical data.

Preliminary efficacy was determined using inferential statistics and effect sizes. Since the data for many outcomes were not normally distributed (Appendix A—outcome measures histograms), Wilcoxon signed-rank tests were used to compare data at baseline and 6 weeks. Effect sizes (Wilcoxon r) were calculated using the formula: Z divided by the square root of n and interpreted as a small effect (Wilcoxon r = 0.1 to <0.3), medium effect (Wilcoxon r = 0.3 to <0.5) or large effect (Wilcoxon r ≥ 0.5). An alpha value of *p* < 0.05 was used to denote statistical significance [46].

## 3. Results

### 3.1. Recruitment and Retention

Thirty-four potential participants were screened, with twenty-eight (82%) assessed as eligible. Of those eligible, 21 (75%) consented to participate in the study (18 Lokomat^®^, 1 Armeo^®^Spring, 2 Armeo^®^Spring and Lokomat^®^). Recruitment continued until the study period was complete (30 June 2021) and the target sample size of 20 was reached, with *n* = 21 ultimately enrolled. Sixteen out of twenty-one participants (Lokomat^®^ *n* = 15, Armeo^®^Spring *n* = 3, two participants completed Armeo^®^Spring and Lokomat^®^) completed the six-week follow-up assessments (Figure 1).

### 3.2. Participant Characteristics

Participant baseline demographic and diagnostic characteristics are provided in Table 1. The mean age was 15.7 ± 4.9 years, 57% were female, and the mean Body Mass Index (BMI) was 21.6 ± 6.0 kg/m^2^. The most common tumour type was medulloblastoma (23%), followed by sarcoma (19%) and Hodgkin lymphoma (19%), with 81% of participants undergoing level 3 (very intensive cancer therapy) or above therapy.

### 3.3. Safety

Over the course of the Lokomat^®^ and Armeo^®^Spring training program, 49 adverse events were recorded (Table 2). The majority (92%) of adverse events were mild (e.g., delayed onset muscle soreness or joint pain). Two moderate adverse events were deemed to be “certainly related” to robotics training (left lower limb nerve pain and left hip pain). One severe adverse event was recorded; however, this was considered unlikely to be related to robotics training (a fractured wrist from a fall at home). There were no grade 4 or 5 adverse events.

In response to the adverse events, exercise sessions were uninterrupted in thirty-four instances, modified in eight instances and stopped in seven instances. No adverse events led to withdrawal from the study. Nearly all participants (*n* = 19) recorded at least one adverse event (with two participants experiencing seven events each).

Common toxicities of cancer therapy were reported in eight participants (total instances *n* = 19). In response to these toxicities, exercise sessions were stopped in two instances, modified in three instances and uninterrupted in the remaining fourteen instances. Toxicities included impaired renal function, ocular graft-versus-host disease affecting vision, recent blood transfusions, lumbar puncture/bone marrow biopsy pain, thrombocytopenia, neuropathic pain related to cancer surgery, anaemia, acute pericarditis, pre-syncopal episodes, low blood sugar, chemotherapy induced nausea/lethargy and epistaxis (platelet count 171 × 10^9^/L).

### 3.4. Feasibility

#### 3.4.1. Demand

A total of 34 paediatric and AYA cancer patients were assessed for eligibility: 21 consented to study, 7 declined and 6 did not meet the inclusion criteria. Reasons for declined participation included: consented to study and withdrew prior to baseline assessment due to extended COVID-19 lockdown (*n* = 2), declined participation no reason given (*n* = 2), and approached and unresponsive to invitation (*n* = 3). Demand for Armeo^®^Spring was poor with only three participants consenting to Armeo^®^Spring intervention.

#### 3.4.2. Adherence

Sixteen of twenty-one participants (76%) completed the six-week intervention, with five withdrawals from the Lokomat^®^ intervention and no withdrawals from the Armeo^®^Spring intervention. Reasons for withdrawal included: relapsed disease (*n* = 2), not challenging enough (*n* = 1), disinterested (*n* = 1), and completed 11 of 12 sessions but not the 6-week assessment due to COVID-19 lockdown (*n* = 1). Overall adherence to the robotics rehabilitation was 83% (231 robotics sessions attended of the 276 prescribed).

#### 3.4.3. Frequency of Needing to Interrupt or Cease Sessions

Mechanical issues with the Lokomat^®^ occurred in eight of the two hundred thirty-one sessions (3%), with seven mechanical issues causing the sessions to be interrupted and one mechanical issue resulting in the session being ceased. A technician was required to troubleshoot and repair the Lokomat^®^ on three occasions. No mechanical issues were recorded with the Armeo^®^Spring.

#### 3.4.4. Patient-Reported Outcomes

Sixteen of the twenty-one participants completed the six-week assessment, including the feedback questionnaire. The majority reported that robotics training sessions were fun (15/16), comfortable (14/16) and that they liked coming (13/16). Fifteen participants reported perceived improvement. Participants generally agreed that the frequency (15/16) and duration (16/16) of sessions were appropriate. Twelve participants noted that they would continue robotics-based training if it were available (Table 3).

Participants commented that the best parts of Lokomat^®^ and Armeo^®^Spring training were their improvement (*n* = 9) and the motivation provided by the games (*n* = 6). Participants commented that the worst part of robotics was finding sessions hard and tiring (*n* = 4), the lack of opportunity to continue (*n* = 2) and the travel/school interruption (*n* = 2). The overall rating given to robotics by participants was an average of 8.8/10.

One hundred and three Canadian Occupational Performance Measure problem areas were identified. 87/103 (84%) of the goals were activity level goals (e.g., Improve meal preparation, dress/toilet independently, get off the floor independently) and 16/103 (16%) were participation level goals (e.g., play baseball, increase social interaction with friends, participate in physical education classes at school).

#### 3.4.5. Efficacy

Preliminary efficacy analyses on Lokomat^®^ intervention (Table 4) suggested that Lokomat^®^ training had substantial beneficial effects on VO_2peak_ (Wilcoxon r = 0.63), 10 m comfortable pace walk (r = 0.51) and maximal pace walk (r = 0.60), 6-min walk test (r = 0.60), maximal back and leg strength (r = 0.71), trunk flexibility (r = 0.60), EORTC QLQ C30 (r = 0.61), FACIT Fatigue r = 0.53 and the Canadian occupational performance measure satisfaction (r = 0.88) and performance scores (r = 0.83) and moderate beneficial effects on LSI (r = 0.30). Due to the very small sample size for the Armeo^®^Spring intervention (*n* = 3), we calculated pre-post effect sizes, but did not undertake inferential analyses (Appendix A—Preliminary efficacy results-Armeo^®^Spring). Results were suggestive of large improvements (i.e., r > 0.5) for most outcomes, including arm range of motion, strength, activities of daily living, fatigue, physical activity and quality of life.

## 4. Discussion

This study aimed to determine the safety, feasibility, and preliminary efficacy of upper and lower limb robot-based therapy for paediatric and AYA cancer survivors. Results suggested that Lokomat^®^ and Armeo^®^Spring training was safe and feasible, and preliminary efficacy outcomes indicated that robot-based training led to large beneficial effects on VO_2peak_, 10 m comfortable and maximal pace walk, 6-min walk test, maximal back and leg strength, trunk flexibility, EORTC QLQ C30, FACIT Fatigue and the Canadian Occupational Performance Measure satisfaction and performance scores.

Lokomat^®^ and Armeo^®^Spring training was safe, with no medically significant grade 3 or above adverse events relating to robot-based therapy recorded and no withdrawals relating to adverse events. Participants did, however, report a number of grade 1 “minor” and grade 2 “moderate” adverse events relating to Lokomat^®^ and Armeo^®^Spring intervention. The most common “adverse events” related to joint pain and delayed onset muscle soreness, which is consistent with other robot assisted gait training studies [26]. While there were a number of grade 1 and 2 adverse events, they were unlikely to limit participation, and patients were often able to continue training sessions without modification of the intensity, duration or frequency of intervention. Unfortunately, a small subset (*n* = 3) of participants with severe dysfunction relating to pelvic surgery and/or neuromuscular conditions experienced multiple episodes of pain requiring regular modification of the harness, rest periods between sessions and adjustment of Lokomat^®^ parameters. While these particular participants had multiple episodes of pain and had severe functional dysfunction related to their cancer and or cancer therapy, they also demonstrated the greatest improvements from robot-based therapy, highlighting the need for better documentation of adverse events. Bessler and colleagues [26] (p. 1–17) suggest that “there is a need for more structured and complete recording and dissemination of adverse events related to robotic gait training to increase knowledge on risks”, with studies often failing to report on the causality and outcome of the adverse event. We included delayed onset muscle soreness as an adverse event as in severely impaired participants it often limited participation or required adjustment of the parameters of the robot due to the associated discomfort. However, delayed onset muscle soreness could be considered a positive outcome from robotics training as it may be an indication that the musculoskeletal system is adapting to an exercise stimulus. Further, on three occasions Lokomat^®^ and Armeo^®^Spring sessions did not proceed due to common toxicities of cancer therapy and participation was limited on 14 occasions either via reduced exercise intensity or the duration of the training session. We recommend Lokomat^®^ and Armeo^®^Spring sessions are staffed by experienced clinicians with expertise in Lokomat^®^ and Armeo^®^Spring training and oncology to screen for and limit potential musculoskeletal and common toxicity adverse events in this cohort.

Program uptake was strong, and comparable to other paediatric and AYA exercise studies [47]. We noticed that uptake appeared to be biased towards participants with moderate to severe functional deficits (e.g., recent limb surgery, hemiparesis, proximal myopathy, gait deficiency) relating to their cancer diagnosis or treatment, and that the appeal of the robot-based intervention seemed to diminish for potential participants with minor functional deficits and general deconditioning. Interestingly, comparable observations have been reported in stroke and cerebral palsy, indicating that the severity of impairment may influence uptake and outcomes [27]. Similarly, we achieved strong program uptake from patients with tumour-types that are typically underrepresented in the literature on exercise in paediatric and AYA cancer (e.g., Ewing sarcoma, osteosarcoma and brain tumours) [1,47,48]. These tumour types are often associated with moderate to severe functional deficits, as surgical intervention is usually indicated as standard therapy [28,49,50]. The resulting functional deficits may deter or exclude patients from engaging in conventional exercise programmes [1]. As such, it appears that the Lokomat^®^ and Armeo^®^Spring devices, offering targeted intervention for upper and lower limb function, may be particularly appealing to patients with such deficiencies.

Given the small sample size, interpretation of preliminary efficacy results will focus on effect sizes, rather than statistical significance. Results suggested that Lokomat^®^ training may lead to large effects on VO_2peak_, 10 m comfortable and maximal pace walk, 6-min walk test, maximal back and leg strength, trunk flexibility, EORTC QLQ C30, FACIT fatigue, and the Canadian occupational performance measure satisfaction and performance scores with moderate effects on the leisure score index. Similar improvements in functional performance measures such as gait speed (10 MWT) and endurance (6 MWT) have been observed in stroke [18], cerebral palsy [24] and multiple sclerosis [25,51]. Objective measurement of back and leg strength, trunk flexibility, quality of life and fatigue have not been measured using the same methods employed by this study, so comparison is difficult. We have previously [1] investigated the benefits of conventional exercise programmes in AYA cancer patients who recently completed systemic cancer therapy and were unable to demonstrate improvements in back and leg strength, trunk flexibility, quality of life and fatigue. While the study was not powered to assess efficacy, the large effect sizes are noteworthy particularly in VO_2peak_, as a number of participants were undergoing concurrent cancer therapy which would be expected to cause a deterioration in function [52]. Of note, participants in our study were able to demonstrate an improvement in cardiorespiratory fitness (1.5 mL·kg^−1^·min^−1^) despite the majority of participants undergoing the intensive chemotherapy. Conventional chemotherapy is known to decease VO_2peak_ by 1.6 mL·kg^−1^·min^−1^ [52].

A key strength of this study was that it is internationally novel, evaluating the feasibility and preliminary efficacy of Lokomat^®^ and Armeo^®^Spring training for cancer patients. We included a comprehensive battery of safety, feasibility and efficacy outcomes and assessments, wherever possible using high-quality instruments with well-established reliability and validity. In particular, we used the Common Terminology Criteria for Adverse Events v5.0 to grade and report adverse events and built a purpose-designed screening tool to screen participants prior to each robot-based therapy session. Further we considered the broader clinical picture (e.g., cancer toxicity) which allowed us to glean a detailed picture of the safety issues regarding robotics training for this complex population.

Limitations must also be considered. Most importantly, the sample size was modest, and a pre-post research design was used. As such, it is unclear how widely the findings can be generalised (e.g., in terms of age- and tumour subgroups). In addition, as a pre-post study, each participant served as their own control. The lack of a high-quality control group means that the efficacy results must be interpreted with caution, as it is unclear how much of the observed improvement was due to the Lokomat^®^ and Armeo^®^Spring training alone.

Assessment of efficacy was also limited by suboptimal uptake of outcome measures including cardiopulmonary fitness testing, maximal back and leg strength and trunk flexibility in the Lokomat^®^ group. This was attributed to weight-bearing restrictions or an inability to perform the outcome measure due to the presence of ambulatory conditions, poor balance or pain. Armeo^®^Spring preliminary efficacy results are presented in Appendix A and will not be interpreted due to the low uptake and small sample size (*n* = 3).

Furthermore, whilst we have presented program uptake data, this was only based on patients who were referred to the program. We are anecdotally aware that surgeons were reluctant to refer patients who had undergone recent limb salvage surgery, due to the lack of safety and efficacy evidence and perceived risk of injury. Therefore, whilst safety results from this study were promising, the study did not include the most complex clinical presentations.

## 5. Conclusions

Lokomat^®^ and Armeo^®^Spring training appears to offer some key advantages for cancer patients with moderate to severe functional limitations. Specifically, it may provide a pathway into exercise training for patients who may be unable (e.g., due to limited functional capacity) or unwilling (e.g., due to issues relating to body image) to access more conventional exercise avenues. Despite Lokomat^®^ and Armeo^®^Spring intervention being hospital-based, intensive, and targeting impairment level dysfunction, it actually assisted our participants to function and participate better in their activities of daily living. In addition, the robotics platform incorporates gamification, which may enhance motivation (both during a session and over time) and may offer a distraction from the physical effort required to engage in exercise.

Some practical barriers were encountered which are important to consider for the ongoing implementation of Lokomat^®^ and Armeo^®^Spring-based training. In particular, the study staff who facilitated the training sessions had specialised knowledge of exercise in oncology, and training in the use of the Lokomat^®^ and Armeo^®^Spring devices. On occasions when study staff were unavailable, it was difficult to find appropriately trained staff to conduct the training sessions. If Lokomat^®^ and Armeo^®^Spring training is to be offered on an ongoing basis, additional training will be needed to share this knowledge and skills more widely with other hospital staff. Furthermore, it is important to acknowledge that Lokomat^®^ and Armeo^®^Spring based training is time and resource-intensive. In this study, two staff were present during training sessions, which was necessary for equipment set-up and session safety. Thus, for robot-based training to be offered as a clinical service, it will be important for departments to devote a budget to both the initial gym set-up, and ongoing staff resources to deliver the training sessions.

Based on this study’s positive safety, feasibility and preliminary efficacy results, it appears that future, larger-scale research is warranted to more rigorously evaluate the efficacy of robotics-based training in young cancer patients and to optimise training regimens. The preliminary efficacy results provided by this study may be used to inform sample size calculations. Such a trial will likely need to be multi-site to achieve a sufficient sample size. This would also improve the generalisability of study findings.

In conclusion, our results suggest that Lokomat^®^ and Armeo^®^Spring rehabilitation is safe and feasible for use in children and AYAs who are currently undergoing or recently completed cancer therapy. A larger controlled trial investigating the efficacy of robotics rehabilitation in this cohort is warranted.

## Figures and Tables

**Figure 1 cancers-15-01250-f001:**
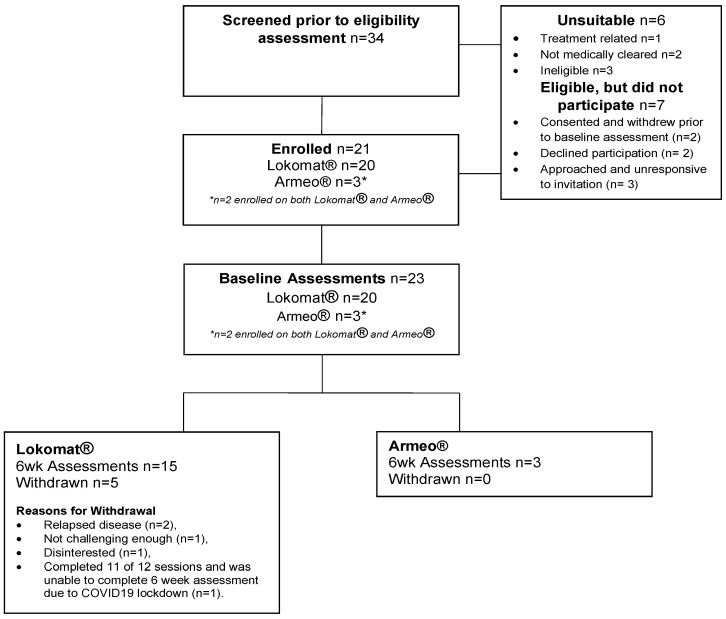
Participant Flow.

**Table 1 cancers-15-01250-t001:** Baseline Characteristics of Participants.

Baseline Characteristics of Participants (*n* = 21)			
Variable	No.		%
Age mean ± Standard Deviation (SD), year (y)		16 ± 5	
Female%	12		57
Weight, mean ± SD, kg		54 ± 22	
Body Mass Index (BMI), mean ± SD, kg/m^2^		21 ± 8	
Diagnosis			
Sarcoma			19
Ewing Sarcoma	3		
Osteosarcoma	1		
Leukaemia			14
* Therapy-related Acute Myeloid Leukaemia	1		
Pre-B Acute Lymphoblastic Leukaemia	2		
Lymphoma			19
Classical Hodgkin Lymphoma	3		
* Burkitts Lymphoma	1		
Germ Cell			
Central Nervous System Germ Cell Tumour	1		4
Other			
Medulloblastoma	5		23
Pilocytic Astrocytoma	3		14
Sacral meningioma-papillary variant, G3	1		4
Myxopapillary Ependymoma	1		4
Treatment Intensity			
Level 1 Least Intensive	1		5
Level 2 Moderately Intensive	3		14
Level 3 Very Intensive	14		67
Level 4 Most Intensive	3		14

* Secondary cancer diagnosis.

**Table 2 cancers-15-01250-t002:** Grade and causality of adverse events (*n* = 49).

		Causality		
		Unlikely to be Related	Possibly/Probably Related	Certainly Related
Grade	1—mild, intervention not indicated	16	27	2
	2—moderate, minimal intervention	0	1	2
	3—severe or medically significant	1	0	0
	4—life threatening consequences	0	0	0
	5—death related to adverse event	0	0	0

Armeo^®^Spring adverse events *n* = 5 (Grade, 1 mild intervention not indicated *n* = 5, Causality, Unlikely to be related *n* = 1, Possibly/Probably related *n* = 4).

**Table 3 cancers-15-01250-t003:** Summary of responses to patient-reported questionnaire (*n* = 16).

	Strongly Agree	Agree	Neutral	Disagree	Strongly Disagree
Sessions were fun	8	7	1	0	0
I was comfortable	8	6	2	0	0
I liked coming	8	5	3	0	0
I saw improvement	11	4	1	0	0
Continue if available	6	6	4	0	0
Freq appropriate	6	9	1	0	0
Duration appropriate	6	10	0	0	0

**Table 4 cancers-15-01250-t004:** Efficacy results in the Robot Oncology Study.

Lokomat					
Outcome Measure	Baseline All Participants Mean SD n	Baseline Completers Mean SD n	6 Weeks Mean SD n	Effect Size	*p* Value
VO_2Peak_ (mL·kg^−1^·min^−1^)	22.63 (9.1) *n* = 10	21.8 (6.0) *n* = 7	23.3 (6.8) *n* = 7	0.63	0.018
10 MWT Comfortable (s)	9.7 (1.7) *n* = 19	9.9 (1.6) *n* = 14	7.7 (2.0) *n* = 14	0.51	0.004
10 MWT Maximal (s)	7.0 (1.7) *n* = 19	7.5 (1.3) *n* = 14	5.3 (3.4) *n* = 14	0.60	0.001
6 MWT(m)	416.0 (174.3) *n* = 19	419.1 (145.0) *n* = 14	487.29 (158.9) *n* = 14	0.60	0.002
Max Back and Leg Strength	44.4 (46.2) *n* = 8	28.0 (16.7) *n* = 5	40.8 (16.0) *n* = 5	0.71	0.043
Trunk Flexibility	12.25 (10.9) *n* = 20	15.1 (6.6) *n* = 10	19.2 (3.6) *n* = 10	0.60	0.059
EORTC QLQC30 *	63.8 (15.1) *n* = 20	63.3 (13.3) *n* = 15	73.3 (19.7) *n* = 15	0.61	0.018
FACIT fatigue **	32.8 (10.4) *n* = 20	31.6 (9.5) *n* = 13	37.5 (10.6) *n* = 15	0.53	0.039
GLTPAQ–LSI ***	27.8 (31.0) *n* = 20	25.3 (22.4) *n* = 15	36.0 (30.6) *n* = 15	0.30	0.249
COPM Performance ****	3.9 (1.1) *n* = 19	3.7 (1.0) *n* = 15	6.3 (1.9) *n* = 15	0.83	0.001
COPM Satisfaction	3.3 (1.4) *n* = 19	3.3 (0.9) *n* = 15	6.6 (1.7) *n* = 15	0.88	0.001

* European Organisation for Research and Treatment of Cancer, Quality of Life Questionnaire, Global Health Status. ** The Functional Assessment of Chronic Illness Therapy—Fatigue Scale. *** Godin Leisure Time Physical Activity Questionnaire—Leisure Score Index. **** Canadian Occupational Performance Measure.

## Data Availability

The data that support the findings of this study are available on request from the corresponding author. The data are not publicly available due to privacy or ethical restrictions.

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
