# Peer review of "Safety, Feasibility and Efficacy of Lokomat^®^ and Armeo^®^Spring Training in Deconditioned Paediatric, Adolescent and Young Adult Cancer Patients"

_cancers, 2023, doi:10.3390/cancers15041250_

Round 1

Reviewer 1 Report

The text is clear. The main question addressed by the research is relevant and interesting.

This study aim to assess the safety, feasibility and efficacy of rehabilitation robotics in children, adolescents and young adults (AYA) during or soon after cancer treatment. Cancer patients with significant musculoskeletal, neurological, gait and/or upper limb deficiency were recruited.

The conclusions are consistent with the evidence and arguments presented. They address the main question posed, but the discussion section needs to be improved. There is not clearly presented the contribution of this manuscript compared to already obtained results (please you can include references compared to which you claim that the obtained results by the new proposed methods are better).

Author Response

Reviewer 1:

Comment 1.1:

The text is clear. The main question addressed by the research is relevant and interesting. This study aim to assess the safety, feasibility and efficacy of rehabilitation robotics in children, adolescents and young adults (AYA) during or soon after cancer treatment. Cancer patients with significant musculoskeletal, neurological, gait and/or upper limb deficiency were recruited.

The conclusions are consistent with the evidence and arguments presented. They address the main question posed, but the discussion section needs to be improved. There is not clearly presented the contribution of this manuscript compared to already obtained results (please you can include references compared to which you claim that the obtained results by the new proposed methods are better).

Response 1.1: Thanks for these comments. We have rewritten the introduction to clarify the contribution of our study relative to the existing literature (line 101-122):

Studies investigating both the Lokomat® and Armeo®Spring have proved effec-tive and well-tolerated in children and AYA with neuromuscular conditions, including stroke, children with cerebral palsy, traumatic brain injuries, and following orthopae-dic procedures [18, 23-26]. Potential benefits include improvements in gait speed (10MWT) and endurance (6MWT), joint range of movement (upper and lower limb), fatigue and functional performance measures such as the Canadian Occupational Performance Measure. While most studies investigating Lokomat® and Armeo®Spring deem them to be effective, they are generally not superior to conven-tional therapies [27]. There is also a need for greater clarity for clinicians to safely ap-ply the technology [23] as little is known about the causality of adverse events and tol-erability and acceptability of applying this technology [26].    

Despite the progressive development of rehabilitation robotic devices and ro-bot-based therapy, their application to the paediatric and AYA population is still scarce. Children and AYA with cancer suffer from loss of mobility and disability which can limit activities of daily living and compromise quality of life well into survivorship [28]. However, to our knowledge robot-based therapies such as the Lokomat® and Armeo®Spring have not been investigated in children and AYA with cancer. There-fore, the primary aim of this study was to determine the safety, and feasibility of Lokomat® and Armeo®Spring training in children and AYA who were undergoing or recently completed cancer treatment. The secondary aim of this study was to deter-mine whether Lokomat® and Armeo®Spring improved exercise and functional capac-ity, daily performance, quality of life and fatigue.

In addition, we have expanded the section of the discussion comparing results from the current study to previous literature, including adding 6 new references (line 409-423):

Similar improvements in functional performance measures such as gait speed (10MWT) and endurance (6MWT) have been observed in stroke [18], cerebral palsy [24] and multiple sclerosis [25, 51]. Objective measurement of back and leg strength, trunk flexibility, quality of life and fatigue have not been measured using the same methods employed by this study, so comparison is difficult. We have previously [1] investigated the benefits of conventional exercise programmes in AYA cancer patients who recently completed systemic cancer therapy and were unable to demonstrate im-provements in back and leg strength, trunk flexibility, quality of life and fatigue. While the study was not powered to assess efficacy, the large effect sizes are noteworthy par-ticularly in VO2peak as a number of participants were undergoing concurrent cancer therapy which would be expected to cause a deterioration in function [52] Of note, participants in our study were able to demonstrate an improvement in cardiorespira-tory fitness (1.5ml/kg/min-1) despite the majority of participants undergoing the inten-sive chemotherapy. Conventional chemotherapy is known to decease VO2peak by 1.6ml/kg.min-1 [52].

New references:

- Baronchelli, F.; Zucchella, C.; Serrao, M.; Intiso, D.; Bartolo, M.; The Effect of Robotic Assisted Gait Training With Lokomat® on Balance Control After Stroke: Systematic Review and Meta-Analysis. Front. Neurol. 2021 12:661815. doi: 10.3389/fneur.2021.661815

-Conner, B.C.; Remec, N.M.; Lerner, Z.F.; Is robotic gait training effective for individuals with cerebral palsy?A systematic review and metaanalysis of randomized controlled trials Clinical Rehabilitation clinical rehabilitation, 1–10 (2022). DOI: 10.1177/02692155221087084.

-Calabro, R.S.; Cassio, A.; Mazzoli, D.; Andrenelli, E.; Eizzarini, E.; Campanini, I.; Carmignano, S.M.; Cerulli, S.; Chisari, C.; Colombo, V.; et.al. What does evidence tell us about the use of gait robotic devices in patients with multiple sclerosis? a com-prehensive systematic review on functional outcomes and clinical recommendations European Journal of Physical and Reha-bilitation Medicine 2021 October;57(5):841-9DOI: 10.23736/S1973-9087.21.06915-X

-Yeha, S.W.; Linc, L.W.; Tamg, K.W.; Tsaik, C.P.; Hong, C.H.; Kuan, Y.C.; Efficacy of robot-assisted gait training in multiple sclerosis: A systematic review and meta-analysis. Multiple Sclerosis and Related Disorders 41 (2020) 102034. https://doi.org/10.1016/j.msard.2020.102034

-Jones, L.W.; Dolinsky, V.; Haykowsky, M.; Patterson, I.; Allen, J.; Scott, J.M.; Rogan, K.; Khouri, M.; Hornsby, W.; Young, M.; et al. Effects of aerobic training to improve cardiovascular function and prevent cardiac remodeling after cytotoxic therapy in early breast cancer [abstract]. Proc. American Association for Cancer Research 102nd Annual Meeting; 2011. p. a5024

-Ness, K.K.; Hudson, M.M.; Ginsberg, J.P.; Nagarajan, R.; Kaste, S.C.; Marina, N.; Whitton, J.; Robison, L.L.; Gurney, J.G.; Physical Performance Limitations in the Childhood Cancer Survivor Study Cohort (2009) J Clin Oncol 27:2382-2389. DOI: 10.1200/JCO.2008.21.1482

Reviewer 2 Report

The paper is well written and verifies the use of two devices: Lokomat and Hocoma, for young cancer patients. In general, the method is sound and robust, I did not find any major issues with the methodology and results sections. 

There are, however some issues that are important and require corrections:

-- The overall limits of the study. Authors in the title, abstract, and discussion discuss "robotics (sic) based rehabilitation" while in fact, they verify two particular devices, one based on a 23-year-old patent (lokomat, https://patentscope.wipo.int/search/en/detail.jsf?docId=WO2000028927)

This severely limits the possible conclusions, and in my opinion, authors cannot based on their study, conclude about any other devices (of which there are plenty) than the two that were selected. This should be discussed in detail. The use of these two particular devices should be stated in the abstract.

-- Lack of detailed background introduction. The authors do not introduce the devices used and prior studies with the devices. The mode of function of the devices is not explained, so, for example, possible adverse effects are hard to imagine.  The authors do not explain, based on current literature, the state of robotic devices for therapy. The current literature is used in a very limited way in the discussion. 

-- I strongly suggest using robot-based therapy instead of robotics, robotics is an engineering discipline I understand that the authors wanted to say that the therapy was based on robots not that the therapy was based on an (abstract) discipline

-- I suggest that authors provide better visualization for the key findings and measures. The authors state that the data was not normally distributed but do not visualize it. Please provide adequate graphs so the readers can understand the distribution

-- Authors explain that there was no control group, however still in the discussion comparison to other interventions should be given. Currently, there is just a statement that "these improvements are noteworthy given that a number of participants ..." without an adequate reference. Particularly as authors discuss feasibility, possible alternatives should be discussed.

-- in the introduction authors state that "there is no published research using these approaches (unclear which approach but I assume rehabilitation robotics) in paediatrics and adolescents. This is not true as Pubmed search shows 22 studies for the keywords lokomat and paediatric. Maybe the authors wanted to state that there are no studies about the use of lokomat for children with cancer but the wording, in this case, has to be improved. Also, changing the keywords to robot therapy cancer gives 2711 results. Particularly as authors in fact study only two particular devices precise explanation of this should be given. 

Overall I think it is a good paper about the possibility of the use of Lokomat and Armeo in young patients' mobility training (possibly fitness) but the current stated scope -- "robot rehabilitation" is in no way supported by the study. Authors should consider greatly narrowing down the scope, preparing a better introduction, discussing using current literature, and better visualization

Author Response

Reviewer 2:

Comment 2.1: The paper is well written and verifies the use of two devices: Lokomat and Hocoma, for young cancer patients. In general, the method is sound and robust, I did not find any major issues with the methodology and results sections. 

There are, however some issues that are important and require corrections.

Response 2.1: thanks for these comments. We have addressed the issues raised below.

Comment 2.2: The overall limits of the study. Authors in the title, abstract, and discussion discuss "robotics (sic) based rehabilitation" while in fact, they verify two particular devices, one based on a 23-year-old patent (lokomat, https://patentscope.wipo.int/search/en/detail.jsf?docId=WO2000028927)

This severely limits the possible conclusions, and in my opinion, authors cannot based on their study, conclude about any other devices (of which there are plenty) than the two that were selected. This should be discussed in detail. The use of these two particular devices should be stated in the abstract.

Response 2.2: Thank you for your comments, we have adjusted the title, abstract, introduction and discussion to narrow the scope of the paper to reflect the two devices (Lokomat® and Armeo®) investigated in this study. 

  • Title: (Page 1, Title, lines 1-4) The title has been adjusted to reflect the two devices investigated (Lokomat® and Armeo®Spring) in this study and not robotics based rehabilitation. The title now reads:

Safety, feasibility and efficacy of Lokomat® and Armeo®Spring training in deconditioned paediatric, adolescent and young adult cancer patients

  • Abstract: Page 1, lines 30-31: … of Lokomat® and Armeo®Spring training in children, adolescents and young adults (AYA) during or soon after cancer treatment.
  • Abstract: Page 2, line 48: … Our results suggest that Lokomat® and Armeo® training is safe and feasible for use in children and AYA who are currently undergoing or recently completed cancer therapy.
  • Discussion: Page 12, lines 454, 458, 465, 467, 469, 472, 483: Throughout the concluding paragraph of the discussion, we have replaced “robotics-based rehabilitation” with “Lokomat® and Armeo®Spring based training” (7 replacements).

Comment 2.3: Lack of detailed background introduction. The authors do not introduce the devices used and prior studies with the devices. The mode of function of the devices is not explained, so, for example, possible adverse effects are hard to imagine.  The authors do not explain, based on current literature, the state of robotic devices for therapy. The current literature is used in a very limited way in the discussion. 

Response 2.3: As requested, we have expanded the background to introduce the concept of rehabilitation robots and included details on the mode and function of both the Lokomat® and Armeo®Spring (Pages 2 and 3 Lines 76-122)

Rehabilitation robots are emerging as a promising therapeutic tool for children and AYA with loss of mobility and disability [17-21]. They help to restore function and mobility and can be classified by their mechanical structure, i.e. end-effectors or exo-skeletons [17]. End-effectors apply forces to the distal segments of limbs, creating a “mechanical chain” that prompts other parts of the limb to move [17]. In contrast, ex-oskeletons are “wearable machines” that are adjusted to the patient’s skeletal structure and move the joint of the limb where the exoskeleton is worn [17].

There are multiple upper and lower limb rehabilitation robotic devices commer-cially available, and or in proof of concept [17]. The Lokomat® (Hocoma AG, Volke-tswil, Switzerland) is a commercially available lower limb robotic exoskeleton with a body weight support system used in parallel with a treadmill that replicates lower limb biomechanics and interfaces with an augmented virtual reality system. Linear electri-cal motors help drive the hip and knee while the foot lift induces passive dorsiflexion of the ankle [17]. Armeo®Spring (Hocoma AG, Volketswil, Switzerland) is a commer-cially available upper limb exoskeleton that supports the participant’s arm and in-cludes an adjustable suspension system which connects with game-based virtual reali-ty with varying degrees of complexity [22]. The exoskeleton can be calibrated to the patient’s active range of movement and provides information about specific move-ment parameters such as resistance, strength, range of motion, and coordination [22].

The use of rehabilitation robots such as the Lokomat® and Armeo®Spring can bring advantages over conventional physiotherapy, as they allow extensive and pre-cise repetition of movement in patients with loss of mobility or disability, require re-duced manual handling for clinicians administering the therapy and provide quanti-tative feedback of range of motion and strength with each repetition. Moreover, their ability to interface with game-based virtual reality may also improve patient motiva-tion and engagement [17, 22].

Studies investigating both the Lokomat® and Armeo®Spring have proved effec-tive and well-tolerated in children and AYA with neuromuscular conditions, including stroke, children with cerebral palsy, traumatic brain injuries, and following orthopae-dic procedures [18, 23-26]. Potential benefits include improvements in gait speed (10MWT) and endurance (6MWT), joint range of movement (upper and lower limb), fatigue and functional performance measures such as the Canadian Occupational Performance Measure. While most studies investigating Lokomat® and Armeo®Spring deem them to be effective, they are generally not superior to conven-tional therapies [27]. There is also a need for greater clarity for clinicians to safely ap-ply the technology [23] as little is known about the causality of adverse events and tol-erability and acceptability of applying this technology [26].    

Despite the progressive development of rehabilitation robotic devices and ro-bot-based therapy, their application to the paediatric and AYA population is still scarce. Children and AYA with cancer suffer from loss of mobility and disability which can limit activities of daily living and compromise quality of life well into survivorship [28]. However, to our knowledge robot-based therapies such as the Lokomat® and Armeo®Spring have not been investigated in children and AYA with cancer. There-fore, the primary aim of this study was to determine the safety, and feasibility of Lokomat® and Armeo®Spring training in children and AYA who were undergoing or recently completed cancer treatment. The secondary aim of this study was to deter-mine whether Lokomat® and Armeo®Spring improved exercise and functional capac-ity, daily performance, quality of life and fatigue.

The comment suggesting that the discussion should provide make further reference to existing literature was also raised by Reviewer 1 (comment 1.1. We have added a paragraph and 6 new references. Please see response 1.1 for full details).

Comment 2.4: I strongly suggest using robot-based therapy instead of robotics, robotics is an engineering discipline I understand that the authors wanted to say that the therapy was based on robots not that the therapy was based on an (abstract) discipline.

Response 2.4: Thanks for this suggestion. We have now changed the terminology throughout the manuscript to “robot-based therapy” or further refined to include the specific device i.e. Lokomat® or Armeo®Spring when applicable. Changes are identified in the manuscript tracked changes document (34 replacements throughout the manuscript). 

 Comment 2.5: I suggest that authors provide better visualization for the key findings and measures. The authors state that the data was not normally distributed but do not visualize it. Please provide adequate graphs so the readers can understand the distribution

Response 2.5: As requested, we have provided histograms in supplementary file 3 to allow readers to see the distribution of the data. A number of the variables are obviously skewed, which is why we used non-parametric tests (as already described in the methods, Line 249).

Since the data for many outcomes were not normally distributed, Wilcoxon signed-rank tests were used to compare data at baseline and 6 weeks.

The key findings are currently presented in table format, providing data on means, ranges, p-values, and effect sizes. We could produce line graphs showing the pre and post values, however that would be duplicating data already provided in Table 4, and would take a lot of space (since it would require 11 separate graphs). We feel that the key findings tables are easy for readers to interpret, and provide more information than could be provided in graphical format. Therefore, we would prefer to keep the main findings in table format. However, we are happy to replace it with graphs if the editor indicates this is preferable.

Comment 2.6: Authors explain that there was no control group, however still in the discussion comparison to other interventions should be given. Currently, there is just a statement that "these improvements are noteworthy given that a number of participants ..." without an adequate reference. Particularly as authors discuss feasibility, possible alternatives should be discussed.

Response 2.6: This is the same suggestion as raised in comments 1.1 and 2.3. Please see response 1.1. 

Response 1.1: Thanks for these comments. We have rewritten the introduction to clarify the contribution of our study relative to the existing literature (line 101-122):

Studies investigating both the Lokomat® and Armeo®Spring have proved effec-tive and well-tolerated in children and AYA with neuromuscular conditions, including stroke, children with cerebral palsy, traumatic brain injuries, and following orthopae-dic procedures [18, 23-26]. Potential benefits include improvements in gait speed (10MWT) and endurance (6MWT), joint range of movement (upper and lower limb), fatigue and functional performance measures such as the Canadian Occupational Performance Measure. While most studies investigating Lokomat® and Armeo®Spring deem them to be effective, they are generally not superior to conven-tional therapies [27]. There is also a need for greater clarity for clinicians to safely ap-ply the technology [23] as little is known about the causality of adverse events and tol-erability and acceptability of applying this technology [26].    

Despite the progressive development of rehabilitation robotic devices and ro-bot-based therapy, their application to the paediatric and AYA population is still scarce. Children and AYA with cancer suffer from loss of mobility and disability which can limit activities of daily living and compromise quality of life well into survivorship [28]. However, to our knowledge robot-based therapies such as the Lokomat® and Armeo®Spring have not been investigated in children and AYA with cancer. There-fore, the primary aim of this study was to determine the safety, and feasibility of Lokomat® and Armeo®Spring training in children and AYA who were undergoing or recently completed cancer treatment. The secondary aim of this study was to deter-mine whether Lokomat® and Armeo®Spring improved exercise and functional capac-ity, daily performance, quality of life and fatigue.

In addition, we have expanded the section of the discussion comparing results from the current study to previous literature, including adding 6 new references (line 409-423):

Similar improvements in functional performance measures such as gait speed (10MWT) and endurance (6MWT) have been observed in stroke [18], cerebral palsy [24] and multiple sclerosis [25, 51]. Objective measurement of back and leg strength, trunk flexibility, quality of life and fatigue have not been measured using the same methods employed by this study, so comparison is difficult. We have previously [1] investigated the benefits of conventional exercise programmes in AYA cancer patients who recently completed systemic cancer therapy and were unable to demonstrate im-provements in back and leg strength, trunk flexibility, quality of life and fatigue. While the study was not powered to assess efficacy, the large effect sizes are noteworthy par-ticularly in VO2peak as a number of participants were undergoing concurrent cancer therapy which would be expected to cause a deterioration in function [52] Of note, participants in our study were able to demonstrate an improvement in cardiorespira-tory fitness (1.5ml/kg/min-1) despite the majority of participants undergoing the inten-sive chemotherapy. Conventional chemotherapy is known to decease VO2peak by 1.6ml/kg.min-1 [52].

New references:

(18)Baronchelli, F.; Zucchella, C.; Serrao, M.; Intiso, D.; Bartolo, M.; The Effect of Robotic Assisted Gait Training With Lokomat® on Balance Control After Stroke: Systematic Review and Meta-Analysis. Front. Neurol. 2021 12:661815. doi: 10.3389/fneur.2021.661815

(24)Conner, B.C.; Remec, N.M.; Lerner, Z.F.; Is robotic gait training effective for individuals with cerebral palsy?A systematic review and metaanalysis of randomized controlled trials Clinical Rehabilitation clinical rehabilitation, 1–10 (2022). DOI: 10.1177/02692155221087084.

(25)Calabro, R.S.; Cassio, A.; Mazzoli, D.; Andrenelli, E.; Eizzarini, E.; Campanini, I.; Carmignano, S.M.; Cerulli, S.; Chisari, C.; Colombo, V.; et.al. What does evidence tell us about the use of gait robotic devices in patients with multiple sclerosis? a com-prehensive systematic review on functional outcomes and clinical recommendations European Journal of Physical and Reha-bilitation Medicine 2021 October;57(5):841-9DOI: 10.23736/S1973-9087.21.06915-X

(51)Yeha, S.W.; Linc, L.W.; Tamg, K.W.; Tsaik, C.P.; Hong, C.H.; Kuan, Y.C.; Efficacy of robot-assisted gait training in multiple sclerosis: A systematic review and meta-analysis. Multiple Sclerosis and Related Disorders 41 (2020) 102034. https://doi.org/10.1016/j.msard.2020.102034

(52)Jones, L.W.; Dolinsky, V.; Haykowsky, M.; Patterson, I.; Allen, J.; Scott, J.M.; Rogan, K.; Khouri, M.; Hornsby, W.; Young, M.; et al. Effects of aerobic training to improve cardiovascular function and prevent cardiac remodeling after cytotoxic therapy in early breast cancer [abstract]. Proc. American Association for Cancer Research 102nd Annual Meeting; 2011. p. a5024

(28)Ness, K.K.; Hudson, M.M.; Ginsberg, J.P.; Nagarajan, R.; Kaste, S.C.; Marina, N.; Whitton, J.; Robison, L.L.; Gurney, J.G.; Physical Performance Limitations in the Childhood Cancer Survivor Study Cohort (2009) J Clin Oncol 27:2382-2389. DOI: 10.1200/JCO.2008.21.1482

Comment  2.7: In the introduction authors state that "there is no published research using these approaches (unclear which approach but I assume rehabilitation robotics) in paediatrics and adolescents. This is not true as Pubmed search shows 22 studies for the keywords lokomat and paediatric. Maybe the authors wanted to state that there are no studies about the use of lokomat for children with cancer but the wording, in this case, has to be improved. Also, changing the keywords to robot therapy cancer gives 2711 results. Particularly as authors in fact study only two particular devices precise explanation of this should be given. 

Response 2.7: We have reworded this sentence to clarify that we are referring to rehabilitation using Lokomat® and Armeo®Spring, and that there are no previous studies of robot-based rehabilitation in children with cancer.

Lines 116-117: “…to our knowledge robot-based therapies such as the Lokomat® and Armeo®Spring have not been investigated in children and AYA with cancer..”

Comment 2.8: Overall I think it is a good paper about the possibility of the use of Lokomat and Armeo in young patients' mobility training (possibly fitness) but the current stated scope -- "robot rehabilitation" is in no way supported by the study. Authors should consider greatly narrowing down the scope, preparing a better introduction, discussing using current literature, and better visualization

Response 2.8: thanks for these comments summarising the comments the reviewer already made regarding the study scope (comment 2.4 and 2.7), introduction (comment 2.3), discussion comparing to existing literature (comment 2.3) and visualisation (comment 2.5). We have undertaken amendments in response to each of these comments, as described in responses 2.3, 2.4, 2.5 and 2.7.